# Very Low-Carbohydrate Ketogenic Diet for the Treatment of Severe Obesity and Associated Non-Alcoholic Fatty Liver Disease: The Role of Sex Differences

**DOI:** 10.3390/nu12092748

**Published:** 2020-09-09

**Authors:** Marco D’Abbondanza, Stefano Ministrini, Giacomo Pucci, Elisa Nulli Migliola, Eva-Edvige Martorelli, Vito Gandolfo, Donatella Siepi, Graziana Lupattelli, Gaetano Vaudo

**Affiliations:** 1Internal Medicine, Department of Medicine, Università degli Studi di Perugia, 06129 Perugia, Italy; marcodabbo@virgilio.it (M.D.); giacomo.pucci@unipg.it (G.P.); elisanullimigliola@yahoo.it (E.N.M.); eva.martorelli@libero.it (E.-E.M.); vito-gandolfo@virgilio.it (V.G.); donatella.siepi@unipg.it (D.S.); graziana.lupattelli@unipg.it (G.L.); gaetano.vaudo@unipg.it (G.V.); 2Internal Medicine, “Santa Maria” University Hospital, Azienda Ospedaliera di Terni, 05100 Terni, Italy

**Keywords:** very low-carbohydrate ketogenic diet, non-alcoholic fatty liver disease, severe obesity, sex differences, body composition

## Abstract

Very low-carbohydrate ketogenic diets (VLCKDs) are an emerging nutritional treatment for severe obesity and are associated with a significant improvement in non-alcoholic fatty liver disease (NAFLD). Little is known about the effect of sex differences on weight loss induced by following a VLCKD. The aim of this study was to investigate the effects of sex differences on weight loss and NAFLD improvement in patients with severe obesity undergoing a VLCKD. Forty-two females and 28 males with severe obesity underwent a 25-day VLCKD. Anthropometric parameters, bioimpedentiometry, degree of liver steatosis measured by ultrasonography, liver function tests, and glucose homeostasis were measured before and after the VLCKD. Males experienced a significantly larger excess body weight loss (EBWL) and a greater reduction in γ-glutamyl transferase (γGT) than females. Dividing the female group by menopausal status, a significant difference between males and pre-menopausal females was found for both EBWL and γGT. No significant difference between groups was observed for improvement in the Edmonton stage or in the degree of steatosis. We conclude that the efficacy of following a VLCKD in severe obesity is affected by sex differences and, for females, by menopausal status. Males seem to experience larger benefits than females in terms of EBWL and NAFLD improvement. These differences are attenuated after menopause, probably because of changes in hormonal profile and body composition.

## 1. Introduction

The current prevalence of obesity is almost three times that of 1975, and in 2016, obesity was estimated to affect over 650 million persons worldwide, with an increasing burden of cardiovascular, metabolic, musculoskeletal, and neoplastic diseases [1]. Sexual dimorphism affects multiple features of obesity, such as body composition, lipid metabolism, endocrine asset, and low-grade chronic inflammation [2]. Since obesity is more prevalent in females than in males, and females show higher propensity than males to look for a treatment for obesity [3,4], likely due to socio-cultural factors, males are generally under-represented in clinical trials. Nonetheless, only a few studies have explored whether sex differences affect weight loss after surgical and non-surgical treatments, achieving conflicting results [5,6].

Non-alcoholic fatty liver disease (NAFLD) represents the most common liver disease in industrialized countries [7]. It is a common complication of metabolic disorders, such as diabetes mellitus, metabolic syndrome, and severe obesity [8]. Interestingly, although females have a higher prevalence of obesity, the prevalence of NAFLD is higher in males and in post-menopausal females [9]. Furthermore, males have a worse prognosis, with a higher risk of progression toward hepatocellular carcinoma [10]. Genetics, sexual hormones, and different inflammatory profiles have been claimed to explain these differences [11].

Very low-carbohydrate ketogenic diets (VLCKDs) are an emerging nutritional treatment for severe obesity, providing a significant, well-tolerated, and rapid loss of body weight [12]. VLCKDs are characterized by a very low carbohydrate intake (<200 kcal/day) in the context of a very low caloric intake (400–800 kcal/day). The aim of VLCKDs is to promote a shift of energy metabolism from carbohydrates to triglycerides, with the formation of ketone bodies (i.e., β-hydroxybutyrate, acetoacetate, and acetone) [13]. This induces a faster weight loss than balanced low-caloric diets, with no fast weight recovery afterward [14]. For this reason, it has been successfully used in subjects with severe obesity, suitable for bariatric surgery, in order to achieve a fast pre-operative weight loss, which has been demonstrated to reduce the risk of pre- and post-operative complications. In particular, it has been hypothesized that the efficacy of following a pre-operative VLCKD is, at least in part, attributable to the reduction in liver volume and degree of steatosis [15,16,17].

In our previous work, we observed that VLCKDs are associated with a reduction in the ultrasonographically measured degree of liver steatosis and γ-glutamyl transferase (γGT) levels [18]. We chose γGT as a non-invasive biomarker of NAFLD because it is a routinely performed, easily accessible, and non-expensive test. The association between γGT levels and the presence and severity of NAFLD is well-established and it was recently proposed as a predictor of cardiovascular risk related to NAFLD [19].

In this preliminary report, we aimed to investigate the effects of sex differences on weight loss and NAFLD improvement in patients with severe obesity undergoing a VLCKD.

## 2. Materials and Methods

We enrolled 73 patients with severe obesity, consecutively referred to the “Multidisciplinary Unit for the Treatment of Obesity” of Università degli Studi di Perugia between 1 October 2016 and 30 April 2019. Preliminary data, extracted from a subpopulation of the patients included in this study, were previously published [14].

The inclusion criteria were: body mass index (BMI) ≥ 40 kg/m^2^ or BMI ≥ 35 kg/m^2^ with obesity-related comorbidities (e.g., metabolic disorders, respiratory disease, cardiovascular disease, severe osteoarthritis, and psycho-social issues); aged between 18 and 65 years.

The exclusion criteria were: absolute contraindications to following a VLCKD, i.e., Type 1 diabetes mellitus (1 patient), severe chronic kidney disease (estimated glomerular filtration rate < 30 mL/min; 1 patient), secondary causes of obesity (1 patient), major psychiatric disorders (0 patients), severe liver failure (Child–Pugh class B or superior; 0 patients), severe heart failure (New York Heart Association class IV; 0 patients), pregnancy (0 patients), chronic HCV and/or HBV infection (0 patients), and self-reported daily alcohol intake > 30 g/day (0 patients).

Three patients met the exclusion criteria and were therefore excluded from the study. The enrolled population was composed of 42 female patients and 28 male patients.

The primary endpoints of the study were the excess of body weight loss (EBWL) and the reduction in γGT. The secondary endpoints were the variations of obesity grade according to the Edmonton Obesity Staging System (EOSS) and the ultrasonographically measured degree of steatosis.

The study was carried out in accordance with the code of ethics of the World Medical Association for human studies (Declaration of Helsinki, 1964). All patients signed an informed consent form in order to voluntarily participate in this study. The research protocol was approved by the local ethics board of Università degli Studi di Perugia and was registered at ClinicalTrials.gov with the registration number NCT03564002.

### 2.1. Definitions

Diabetes was defined according to the 2020 American Diabetes Association guidelines [20]. Both subjects with an established diagnosis of diabetes and subjects diagnosed with diabetes during our evaluation were considered.

Hypertension was defined, according to the 2018 European Society of Cardiology guidelines [21], as the presence of an established diagnosis of hypertension, ongoing anti-hypertensive treatment, or an in-office observation of blood pressure (BP) ≥ 140/90 mmHg. In-office BP was assessed as the mean of three consecutive measurements, with a time lapse of at least 10 min between each measurement.

Menopause was defined as amenorrhea lasting for more than 12 months at the time of enrolment.

### 2.2. Measurements

All subjects were evaluated at enrolment and after 25 days of following a VLCKD. Weight and height were measured using a scale. BMI was calculated as
(1)$$\frac{\mathrm{weight}\;\left(\mathrm{kg}\right)}{\mathrm{height}\;{\left(m\right)}^{2}}$$

Waist circumference was measured by placing a measuring tape on the horizontal plane around the abdomen at the level of the umbilicus, at the end of a normal expiration.

The severity of obesity was classified according to the EOSS [22]. This system is based on an evaluation of the comorbidities and complications of obesity, divided into four domains: obesity-related diseases or risk factors, physical symptoms, psychological symptoms, and global functioning or quality of life. Impairment in each domain is classified according to a scale ranging from 0 (no impairment) to 4 (severe impairment). The patient is classified according to the highest level of impairment in any domain.

The ideal body weight (IBW; kilograms) was calculated according to the Lorentz formula [23]:For males 50 + [0.91 × (height in centimeters − 152.4)];For females 45.5 + [0.91 × (height in centimeters − 152.4)].

Excess of body weight (EBW; kilograms) was calculated as (weight–IBW). A variation in body weight was reported as EBW Loss (EBWL; %), calculated as:(2)$$\frac{{\mathrm{EBW}}_{\mathrm{baseline}}-{\mathrm{EBW}}_{\mathrm{after}\;\mathrm{intervention}}}{{\mathrm{EBW}}_{\mathrm{baseline}}}\times 100$$

Bioimpedentiometry (50 kHz, amplitude 50 mA, Body Composition Analyzer TBF-410GS; Tanita, Tokyo, Japan), with electrodes applied on the plantar surface of both feet, was used to determine fat mass and fat-free mass as absolute values and percentages of body weight.

Abdominal ultrasonography was performed using a 3.5 MHz convex array probe on a commercially available machine (Easote MyLab 50, Esaote, Genoa, Italy).

Liver steatosis was assessed using a semi-quantitative method, based on the presence of three qualitative criteria: Parenchymal hyperechogenicity, compared to kidney cortical echogenicity; posterior beam attenuation with standard settings; blurred visualization of the intrahepatic vessels and diaphragm. The grades were defined as follows: grade 0 = no steatosis; grade 1 = mild steatosis, a slight and diffuse increase in fine echoes in the hepatic parenchyma with normal visualization of the diaphragm and portal vein borders; grade 2 = moderate steatosis, a moderate and diffuse increase in fine echoes with slightly impaired visualization of the diaphragm and portal vein borders; grade 3 = severe steatosis, a marked increase in fine echoes with poor or no visualization of the diaphragm, portal vein borders, or posterior portion of the right lobe [24].

Blood samples was drawn in the morning after a 13-hour fast. The complete blood cell count and biochemical profile were analyzed with commercially available, completely automated analyzers (DxH 800 AU and DxC 700, CoreLab, Beckman Coulter Italia, Milan, Italy). The measured parameters were glycemia, insulinemia, glycated hemoglobin (HbA1c), aspartate amino-transferase (AST), alanine amino-transferase (ALT), and γGT.

Insulin resistance was determined by using the homeostasis model assessment of insulin resistance (HOMA-IR) with the following equation [25]:(3)$$\text{HOMA-IR}=\frac{\mathrm{fasting}\;\mathrm{serum}\;\mathrm{insulin}\;\left(\frac{\mathrm{mIU}}{\mathrm{mL}}\right)\times \mathrm{fasting}\;\mathrm{plasma}\;\mathrm{glucose}\;\left(\frac{\mathrm{mg}}{\mathrm{dL}}\right)}{405}$$

### 2.3. Very Low-Carbohydrate Ketogenic Diet

Patients underwent a 25-day nutritional treatment with severe caloric restriction (<800 kcal/day). Carbohydrate intake was <50 g/day (corresponding to <200 kcal/day) with a protein intake of 1.4 g/kg of IBW. Assuming an average ideal weight of 60 kg for women and 75 kg for men, the protein intake was >80 g/day (corresponding to >300 kcal/day) for women and >90 g/day for men (corresponding to >350 kcal/day). The remaining caloric intake was composed of fat (<250 kcal/day, corresponding to <30 g/day).

The protein intake was achieved, in part, with a dietary supplement composed of milk whey protein (Nepicomplex, PromoPharma^®^ 39 g/day corresponding to 156 kcal/day) diluted in water or skimmed milk. The remaining part was achieved with fresh aliments in appropriate quantities, depending on ideal body weight (Solus multinutrient and Xalifom, PromoPharma^®^). Details of the diet are shown in Appendix A. The composition of the nutritional supplements was previously described [18].

All patients were also invited to compile a daily diary to make note of any physical symptoms (e.g., headache, constipation, nausea, vomiting, weakness, dizziness, muscle cramps, and palpitations) and hunger. Hunger was self-reported by means of the three-factor eating questionnaire [26]. Water intake was fixed to no less than 2 L/day. Self-testing of urinary ketones was carried out once daily. Urine ketone concentrations were measured using over-the-counter reagent strips (Accu-Chek Ketur Test, Roche Diagnostics GmbH, Mannheim, Germany), which determine the presence of ketones upon reaction with nitroprusside salt. Urine ketones were assessed using a semi-quantitative scale. The compliance of patients to the VLCKD was verified by the presence of ketones in the urine. Patients who did not test positive for urine ketones 96 h after the beginning of the diet, and patients who tested negative during the treatment, were excluded from the study.

### 2.4. Statistical Analysis

Continuous variables are expressed as mean ± standard deviation (SD) when normally distributed. Non-normally distributed variables are expressed as median ± interquartile range (IR). Categorical variables are expressed as numbers (%). The null hypothesis was rejected for a probability (*p*) <0.05 under the null hypothesis assumption. Based on our previous experience [18], we estimated a difference in γGT levels between groups of 12 ± 18 IU/L. For a statistical power of 0.8, a sample size of 35 subjects for each group was requested. Since the male population was under-represented, this work can be considered a “pilot study”.

Comparisons between groups were performed using the Student’s *t*-test and analysis of variance (ANOVA) for normally distributed variables, and the Kruskal–Wallis test for non-normally distributed variables. Correction for multiple comparisons was performed using Fisher’s least significant difference (LSD) test. Comparisons between categorical variables were performed using the χ^2^ test and Fisher’s exact test. Differences in probabilities are expressed as odds ratios (ORs) and 95% confidence intervals (95% CIs). Variations were analyzed through the paired *t*-test and the Wilcoxon test for normally and non-normally distributed variables, respectively. Correlation coefficients between variables were estimated using Pearson’s and Spearman’s correlation tests for normally and non-normally distributed variables, respectively. The effects of potential confounders were removed through partial correlation. Adjustment for potential confounders and independent predictors were analyzed through multivariate linear regression.

## 3. Results

The analyzed population was mainly composed of female subjects (*n* = 42, 60%), 21 of which were post-menopausal with a median age of menopause of 48.5 years (IR = 41–53). All patients had a positive test for urinary ketones up to 72 h after the beginning of the diet. No patient abandoned the diet because of excessive hunger or unacceptable physical symptoms.

The general characteristics of our population according to sex at baseline are reported in Table 1.

No significant difference between groups was observed in BMI, EBW, Edmonton stage, or degree of steatosis. Significant differences were observed in waist circumference, body composition, and γGT levels.

The characteristics of the female population according to menopausal status are reported in Table 2.

Significant differences in terms of glycometabolic status, body composition, AST, ALT, γGT, Edmonton stage, and degree of steatosis were observed between pre- and post-menopausal females, as well as between males and pre-menopausal females, with pre-menopausal women showing a milder degree of steatosis, lower γGT and transaminases levels, and a lower Edmonton stage, irrespective of a worse glycometabolic status. No difference was observed between males and post-menopausal females, except for age and body composition.

A significant correlation was found between γGT levels and the degree of steatosis, as well as Edmonton stage (Figure 1). The correlation between γGT and Edmonton stage was preserved after correction for sex and menopausal status (ρ = 0.387, *p* = 0.003), whereas a correlation between γGT and the degree of steatosis was observed in post-menopausal females only (ρ = 0.563, *p* = 0.015).

After following the VLCKD, a significant weight loss was observed in all groups, with a significant reduction in fat mass and degree of steatosis (Table 3). A significant reduction in Edmonton stage was observed in males and pre-menopausal females only, whereas a significant reduction in γGT, HOMA-IR, and HbA1c was observed for males and post-menopausal females only. No significant reduction in fat-free mass was observed in male subjects.

Comparing our primary outcomes, males experienced a significantly larger EBWL and greater reduction in γGT compared to females (Figure 2). The difference in γGT reduction was confirmed after adjustment for baseline levels of γGT (*p* = 0.003, *p* for interaction = 0.998).

Dividing the female group by menopausal status, the difference among groups in EBWL was borderline significant, with a preserved significant difference between males and pre-menopausal females (Figure 3a). Regarding the reduction in γGT, we found a significant difference among groups, which was lost after correction for baseline γGT levels (*p* = 0.299, *p* for interaction = 0.999). Performing comparisons between single groups, we found significant differences between pre-menopausal females and both males and post-menopausal females (Figure 3b). After adjustment for baseline levels of γGT, only the difference between males and pre-menopausal females remained significant (*p* = 0.022; *p* = 0.088 for post-menopausal females).

In regard to secondary outcomes, no significant differences between groups were observed (Appendix A), although a higher probability of improvement in the Edmonton stage was found for pre-menopausal females and a higher probability of improvement in the degree of steatosis for males.

We also observed a significantly higher reduction in waist circumference in males than in females (8.5 cm; IR = 5.5–11.0 vs. 6.0 cm, IR = 3.0–10.0 cm; *p* = 0.032), as well as a higher reduction in both HOMA-IR and HbA1c in males than in pre-menopausal females (4.57 ± 4.56 vs. 0.44 ± 1.72, *p* = 0.030 for HOMA-IR; 0.40, IR 0.23–1.05 vs. 0.05, IR −0.2 to 0.4, *p* = 0.034 for HbA1c). Performing a linear regression model comprising sex, EBWL, HOMA-IR, and waist circumference, HOMA-IR was the only independent predictor of a reduction in γGT (standardized β = 0.657, *p* < 0.001; *p* of the model <0.001, *R*^2^ = 0.475).

## 4. Discussion

Several biological features differ between males and females, such as body composition, hormonal profile, glycometabolic status, and inflammatory response. As a result, overweight and obesity have different pathological features between males and females. Therefore, weight loss strategies are expected to have different effects on male and female subjects [5]. Despite this, no previous studies have investigated the effect of sex differences on weight loss induced by VLCKDs.

Our results showed that males experience larger benefits than females, particularly if compared to pre-menopausal females, in terms of body weight loss and improvement in NAFLD, evaluated through serum levels of γGT. Males also showed a higher, though non-significant, probability of experiencing an improvement in the ultrasonographically measured degree of steatosis, whereas pre-menopausal females displayed a higher, though non-significant, probability of experiencing an improvement in the Edmonton stage compared to males and post-menopausal females. Conversely, post-menopausal females did not experience a significant improvement in the Edmonton stage, likely because of an average older age.

We can try to explain these differences in light of the hypothesized mechanism of action of VLCKDs, although this is still largely unknown. Basically, the aim of VLCKDs is to promote a metabolic shift from carbohydrates to triglycerides, stored in adipose tissue, as the main energy source for basal metabolism [27]. In particular, visceral adipose tissue is metabolically more active than subcutaneous visceral fat, with enhanced lipolysis and a higher release of free fatty acids. Males have a larger amount visceral fat than females and a higher basal energy expenditure because of higher lean body mass [2]. Both these factors could contribute to the larger benefit of following a VLCKD in males, in terms of body weight loss and NAFLD improvement. Indeed, a reduction in visceral fat is a key factor in the resolution of NAFLD [28]. This hypothesis was confirmed by our observation of a higher reduction in waist circumference in males than in females.

We also observed a higher reduction in both HbA1c and HOMA-IR in males than in pre-menopausal females, supporting the potential role of insulin resistance improvement as well. Visceral fat and insulin resistance are closely connected [29], so differences in insulin improvement could be mediated by different body compositions. However, HOMA-IR is the only independent predictor of a reduction in γGT; thus, other differences in liver metabolism could mediate the relationship between improvements in NAFLD and insulin sensitivity after the VLCKD [30].

Males have a higher probability of developing NAFLD, with a greater chance of experiencing a more severe case of the disease [31]. Hence, an interaction between sex and baseline levels of γGT could be hypothesized. However, differences were confirmed after appropriate correction, without significant interaction.

The use of surrogate endpoints for NAFLD (γGT and ultrasonographically measured degree of steatosis) is undoubtedly the most relevant limitation of our study. The gold standard for the diagnosis and staging of NAFLD is a liver biopsy, but it is an invasive and potentially hazardous procedure. Several biomarkers have been proposed for the noninvasive diagnosis and risk stratification of NAFLD [32], but most of them are still under investigation and none of them have entered common clinical practice yet. The fatty liver index (FLI) is a noninvasive score composed of easily accessible parameters, including γGT, which has good performance in predicting fatty liver disease in general populations of different ages and ethnicities [33]. Unfortunately, FLI has not yet been validated in severely obese subjects, although appropriate studies are expected. Interestingly, in our population, the γGT levels correlated with the Edmonton stage, highlighting the impact of NAFLD and its complications on the quality of life of obese patients. Compared to B-mode ultrasonography, we judged γGT to be more appropriate for quantifying a variation in a short time lapse, because it is a quantitative parameter. Furthermore, the sensitivity of B-mode ultrasonography for the detection of liver steatosis is limited, particularly for mild degrees of steatosis and in subjects with obesity [34]. However, neither γGT nor B-mode ultrasonography can quantify the fat content in the liver, and this is another limitation of our results. Since computed tomography and magnetic resonance imaging can quantify the fat content of the liver, further studies using these techniques are required to confirm our results.

Furthermore, we did not take into account socio-cultural aspects (e.g., education level, income, and social inclusion), which could mediate the relationship between sex and weight loss in severe obesity.

Finally, the small sample size, particularly the under-potentiated male group, and the short follow-up could have prevented us from detecting some significant associations, particularly for the secondary endpoints.

Considering all of these limitations, we can conclude that our study shows, for the first time, that the efficacy of following a VLCKD in severe obesity is affected by sex differences and, for females, by menopausal status. Specifically, males seem to experience larger benefits than pre-menopausal females in terms of EBWL and NAFLD improvement. These differences are attenuated after menopause, probably because of changes in hormonal profile and body composition. Post-menopausal females, instead, were the group who experienced the least benefit from following the VLCKD.

This study highlights the necessity of further studies exploring the effect of sex differences on the efficacy of weight loss strategies for severe obesity, in order to customize treatments based on each patient’s characteristics and to maximize their benefits.

## Figures and Tables

**Figure 1 nutrients-12-02748-f001:**
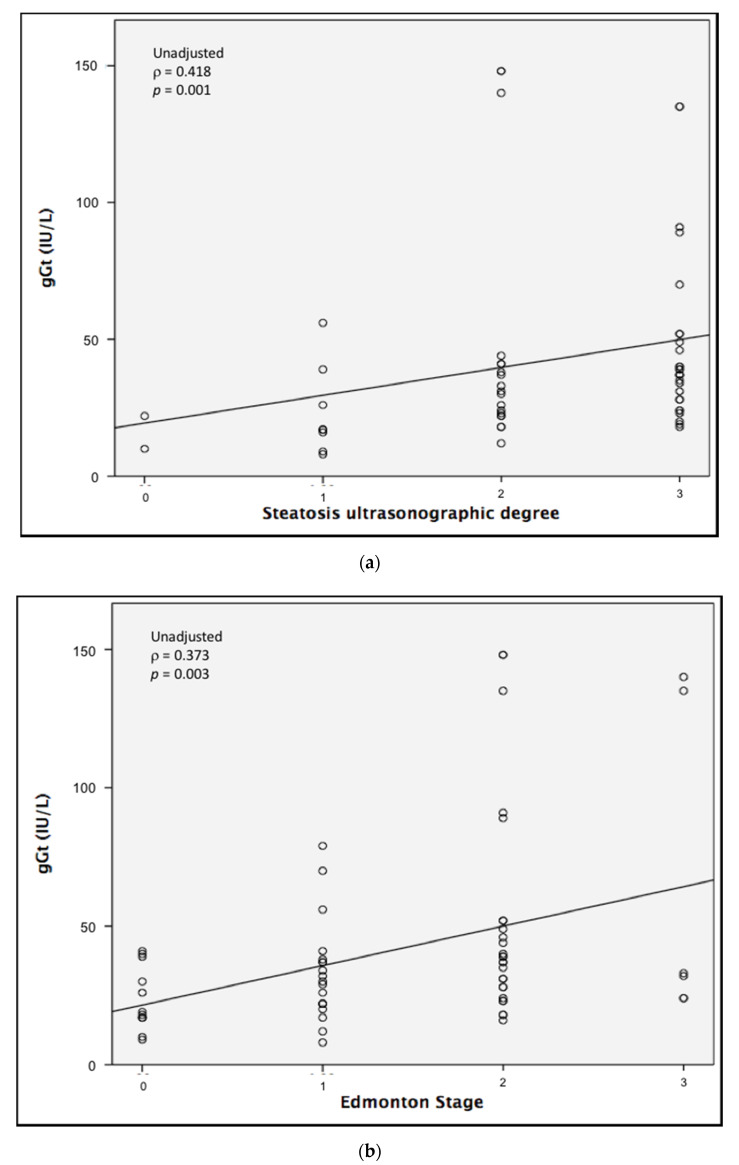
Spearman’s correlation between levels of gamma-glutamyl transferase (gGT) and (**a**) ultrasonographically measured degree of steatosis or (**b**) Edmonton stage.

**Figure 2 nutrients-12-02748-f002:**
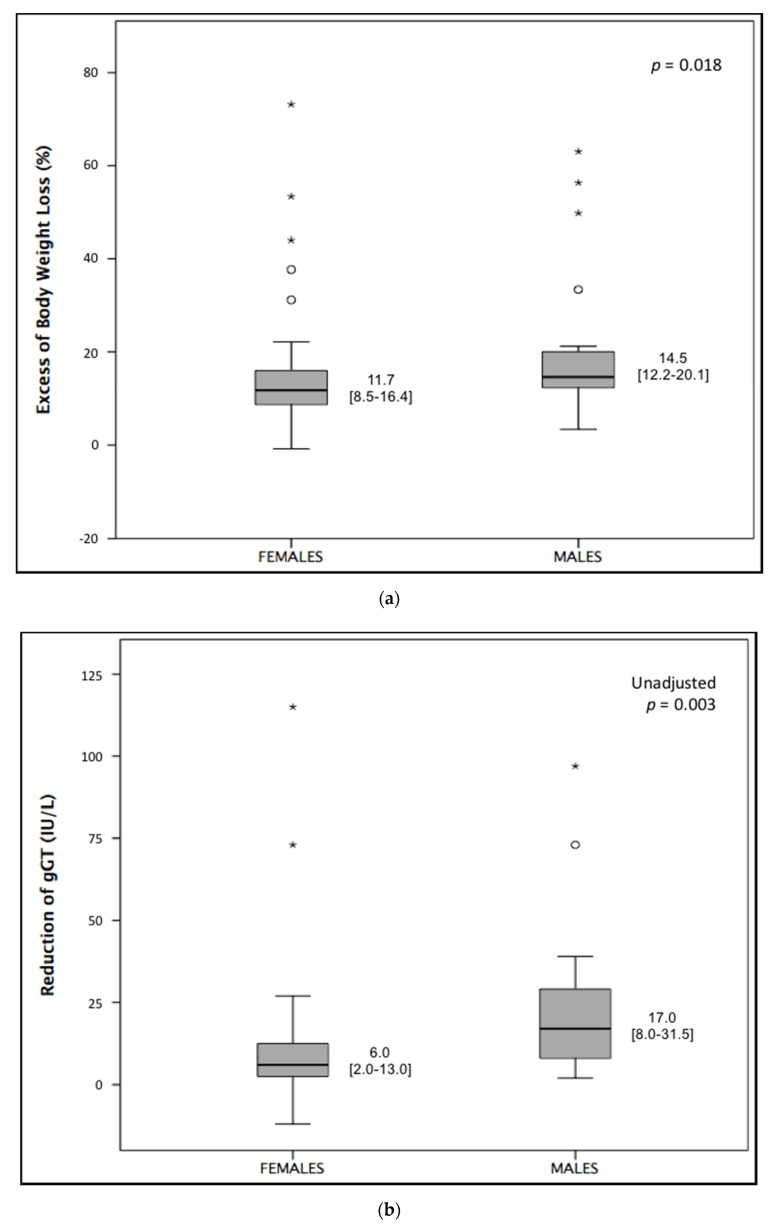
Comparison of (**a**) excess of body weight loss (EBWL) and (**b**) reduction in gamma-glutamyl transferase (gGT) between males and females (Kruskal–Wallis test). * and ◯ depict outliers.

**Figure 3 nutrients-12-02748-f003:**
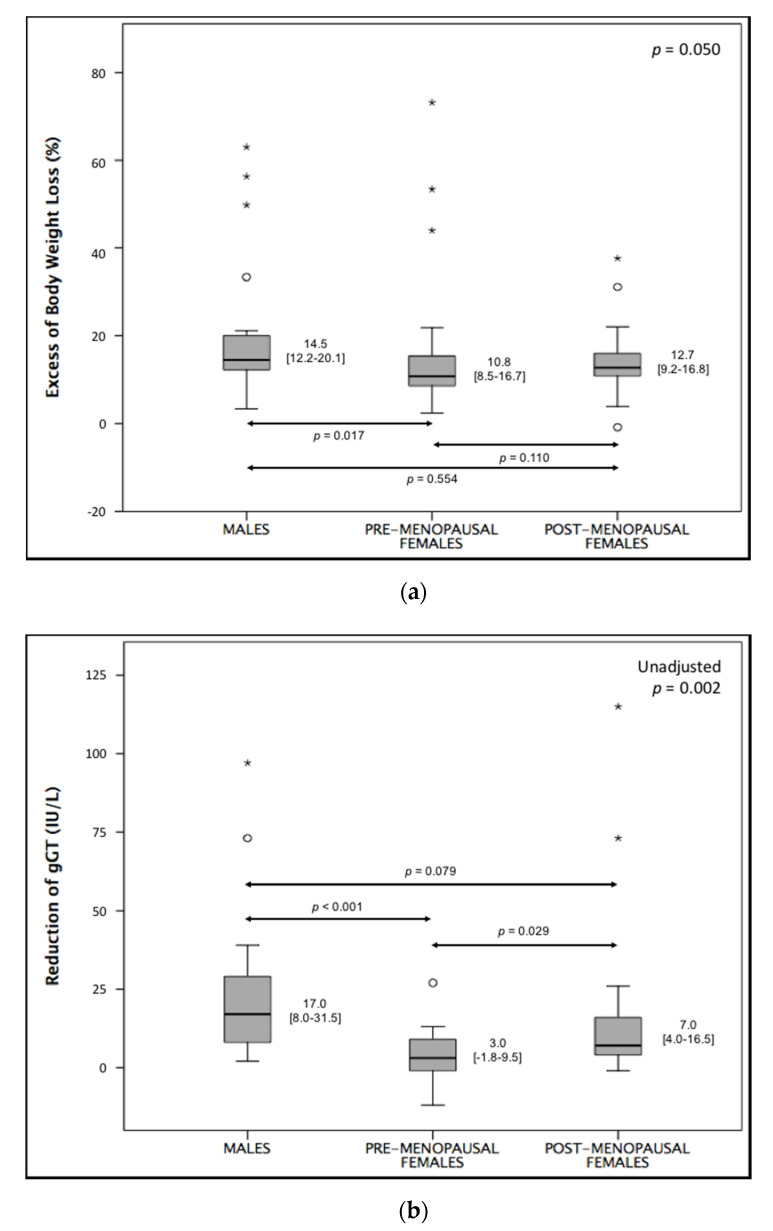
Comparison of (**a**) excess of body weight loss (EBWL) and (**b**) reduction in gamma-glutamyl transferase (gGT) between males, pre-menopausal females, and post-menopausal females (Kruskal–Wallis test). * and ◯ depict outliers.

**Table 1 nutrients-12-02748-t001:** Comparison of the clinical, anthropometric, bioimpedentiometric, biochemical, and ultrasonographic parameters between female and male patients at baseline (Student’s *t*-test, Kruskal–Wallis test, and χ^2^ test).

Parameter		Males (*n* = 28)	Females (*n* = 42)	*p*
Age (years), median (IR)		46 (20–62)	50 (17–67)	0.808
Weight (kg)		148 ± 23	119 ± 20	0.425
Body mass index (kg/m^2^)		48 ± 7	46 ± 8	0.208
Excess of body weight (kg)		70 ± 18	66 ± 21	0.396
Waist circumference (cm)		129 ± 20	140 ± 13	0.008 *
Fasting blood glucose (mg/dL)		120 ± 55	102 ± 23	0.122
HOMA-IR		7.8 ± 6.5	5.5 ± 5.9	0.503
HbA1c (%)		6.5 ±1.5	7.0 ± 5.0	0.342
AST (IU/L)		31 ± 11	24 ± 14	0.448
ALT (IU/L)		46 ± 24	27 ± 16	0.080
γGT (IU/L)		57 ± 38	31 ± 27	<0.001 *
Fat mass (%)		44 ± 9	50 ± 3	<0.001 *
Fat-free mass (kg)		80 ± 14	59 ± 8	<0.001 *
Hypertension, number (%)		15 (55)	19 (45)	0.428
Diabetes mellitus, number (%)		12 (42)	10 (24)	0.405
Active smokers, number (%)		6 (21)	9 (21)	0.621
Liver steatosis, number (%)	No steatosis	0 (0)	2 (5)	0.096
	Grade 1	2 (7)	11 (25)	
	Grade 2	10 (36)	12 (28)	
	Grade 3	16 (57)	17 (40)	
Edmonton stage, number (%)	0	3 (11)	10 (24)	0.308
	1	8 (29)	15 (36)	
	2	14 (50)	14 (33)	
	3	3 (10)	3 (7)	

Significant differences are marked with *. ALT, alanine aminotransferase; AST, aspartate aminotransferase; γGT, gamma glutamyl transferase; HbA1c, glycated hemoglobin; HOMA-IR, homeostasis model assessment of insulin resistance.

**Table 2 nutrients-12-02748-t002:** Comparison of the anthropometric, bioimpedentiometric, biochemical, and ultrasonographic parameters between pre- and post-menopausal aged females at baseline (Student’s *t*-test, Kruskal–Wallis test, and χ^2^ test). Comparisons with male subjects are also reported (for the means and standard deviations (SDs) of male subjects, refer to Table 1).

Parameter		Pre-Menopause (*n* = 21)	Post-Menopause (*n* = 21)	*p*	*p* (Pre-Menopause vs. Males)	*p* (Post-Menopause vs. Males)
Age (years), median (IR)		42 (17–53)	58 (44–67)	<0.001 *	0.206	0.003 *
Weight (kg)		125 ± 18	113 ± 21	0.043 *	0.949	0.569
Body mass index (Kg/m^2^)		47 ± 7	45 ± 9	0.430	0.450	0.176
Excess of body weight (kg)		70 ± 19	61 ± 21	0.546	0.995	0.458
Waist circumference (cm)		127 ± 17	130 ± 19	0.593	0.052	0.171
FBG (mg/dL)		99 ± 23	105 ± 22	0.428	0.038 *	0.467
HOMA-IR		3.9 ± 3.0	7.0 ± 7.0	0.229	0.194	0.983
HbA1c (%)		7.8 ± 9.0	6.0 ± 1.0	0.045 *	0.056	0.284
AST (IU/L)		19 ± 6	30 ± 18	0.014 *	<0.001 *	0.272
ALT (IU/L)		19 ± 5	36 ± 20	0.001 *	<0.001 *	0.509
γGT (IU/L)		23 ± 13	39 ± 34	0.016 *	<0.001 *	0.281
Fat mass (%)		51±4	49 ± 3	0.661	0.001 *	<0.001 *
Fat-free mass (kg)		61 ± 6	57 ± 9	0.114	<0.001 *	<0.001 *
Liver steatosis, number (%)	No steatosis	2 (10)	0 (0)	0.011 *	0.002 *	0.949
	Grade 1	11 (50)	1 (5)			
	Grade 2	4 (20)	7 (33)			
	Grade 3	4 (20)	13 (62)			
Edmonton stage, number (%)	0	7 (33)	3 (14)	0.006 *	0.010 *	0.873
	1	11 (52)	4 (19)			
	2	2 (10)	12 (57)			
	3	1 (5)	2 (10			

Significant differences are marked with *. ALT, alanine aminotransferase; AST, aspartate aminotransferase; γGT, gamma glutamyl transferase; FBG, fasting blood glucose; HbA1c, glycated hemoglobin; HOMA-IR, homeostasis model assessment of insulin resistance; IR, interquartile range.

**Table 3 nutrients-12-02748-t003:** Variations of the clinical, anthropometric, bioimpedentiometric, biochemical, and ultrasonographic parameters before and after following the very low-carbohydrate ketogenic diet (VLCKD), according to sex and menopausal status (Student’s paired *t*-test and Wilcoxon test).

	Males (*n* = 28)				Females (*n* = 42)			Pre-Menopausal Females (*n* = 21)			Post-Menopausal Females (*n* = 21)		
Parameter	Before		After	*p*	Before	After	*p*	Before	After	*p*	Before	After	*p*
Weight (kg)	137 ± 18		124 ± 20	<0.001 *	119 ± 20	109 ± 20	<0.001 *	125 ± 18	113 ± 20	<0.001 *	113 ± 21	104 ± 19	<0.001 *
BMI (kg/m^2^)	46 ± 7		42 ± 7	<0.001 *	46 ± 8	42 ± 8	<0.001 *	47 ± 7	42 ± 8	<0.001 *	45 ± 9	41 ± 8	<0.001 *
Waist (cm)	138 ± 12		128 ± 12	<0.001 *	129 ± 19	121 ± 18	<0.001 *	127 ± 20	118 ± 18	0.003 *	130 ± 18	123 ± 18	<0.001 *
HOMA-IR	11.3 ± 9.2		4.5 ± 4.1	0.004 *	5.5 ± 6.0	2.9 ± 2.8	0.035 *	4.0 ± 3.1	3.4 ± 3.7	0.439	6.6 ± 7.2	2.6 ± 1.8	0.048 *
HbA1c (%)	6.5 ± 1.4		5.9 ± 1.3	<0.001 *	7.1 ± 5.3	5.7 ± 0.7	0.014 *	7.9 ± 8.0	5.5 ± 0.6	0.481	6.4 ± 1.0	5.8 ± 0.8	0.015 *
AST (IU/L)	30 ± 12		35 ± 15	0.153	24 ± 2	24 ± 2	0.956	19 ± 5	20 ± 7	0.456	28 ± 14	27 ± 11	0.815
ALT (IU/L)	47 ± 28		52 ± 38	0.491	26 ± 15	31 ± 17	0.088	19 ± 5	26 ± 18	0.098	33 ± 17	35 ± 16	0.537
γGT (IU/L)	65 ± 42		40 ± 28	<0.001 *	32 ± 30	20 ± 13	<0.001 *	23 ± 14	19 ± 12	0.152	40 ± 37	21 ± 13	<0.001 *
Fat mass (%)	44 ± 9		40 ± 7	0.016 *	50 ± 5	47 ± 5	<0.001 *	50 ± 3	47 ± 7	0.024 *	49 ± 3	48 ± 3	<0.001 *
FFM (kg)	77 ± 14		74 ± 10	0.053	59 ± 8	56 ± 9	<0.001 *	61 ± 6	59 ± 10	<0.001 *	57 ± 9	54 ± 8	<0.001 *
Steatosis grade (%)	0	0.0	6.3	0.002 *	5.6	15.2	<0.001 *	11.1	33.3	0.020 *	0.0	0.0	0.009 *
	1	4.2	37.5		25.0	45.5		44.4	53.3		5.6	38.9	
	2	37.5	37.5		27.8	21.2		22.2	13.3		33.3	27.8	
	3	58.3	18.8		41.7	18.2		22.2	0.0		61.1	33.3	
Edmonton stage (%)	0	10.7	6.7	<0.001 *	23.8	41.9	<0.001 *	33.3	66.7	0.020 *	14.3	18.8	0.083
	1	28.6	20.0		35.7	22.6		52.4	26.7		19.0	18.8	
	2	50.0	53.3		33.3	32.3		9.5	6.7		57.1	56.3	
	3	10.7	20.0		7.1	3.2		4.8	0.0		9.5	6.3	

Significant variations are marked with *. ALT, alanine aminotransferase; AST, aspartate aminotransferase; BMI, body mass index; FFM, fat-free mass; γGT, gamma glutamyl transferase; Hb1Ac, glycated hemoglobin; HOMA-IR, homeostasis model assessment of insulin resistance.

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
