# Peer review of "Very Low-Carbohydrate Ketogenic Diet for the Treatment of Severe Obesity and Associated Non-Alcoholic Fatty Liver Disease: The Role of Sex Differences"

_nutrients, 2020, doi:10.3390/nu12092748_

Round 1

Reviewer 1 Report

The issue is important. In the treatment of severe obesity, sex dimorphism is clinically evident but not considered in clinical guidelines. Response to diet appears to be different between sexes but limitations, correctly stated by the authors are probably too evident. Not only the sample size but also in NAFLD assessment. I detect inappropriate auto-plagiarism with a previous report. Then, the report must indicate both that this is a post-hoc analysis, and more important that this is a very preliminary report. The diet may be supplementary but not the current supplementary table 1, if these data have not been previously published. Figures must be improved. The use of 2 decimals and coma instead of period is unacceptable

Author Response

Comment #1

The issue is important. In the treatment of severe obesity, sex dimorphism is clinically evident but not considered in clinical guidelines. Response to diet appears to be different between sexes but limitations, correctly stated by the authors are probably too evident. Not only the sample size but also in NAFLD assessment.

Response #1

Thank you for your favorable comment.

Comment #2

I detect inappropriate auto-plagiarism with a previous report. Then, the report must indicate both that this is a post-hoc analysis, and more important that this is a very preliminary report.

Response #2

We were aware of potential auto-plagiarism, because a sub-analysis of these data has been previously published. In order to overcome this issue, we removed the description of dietary supplements, referring to the previously published article. More, we declare that "Preliminary data, extracted from a subpopulation of patients included in this study, have been previously published [14]" in the methods section (Page 2, Lines 58-59). We are also stressing the fact of this work being a preliminary report (Lines 52 and 175).

Comment #3

The diet may be supplementary but not the current supplementary table 1, if these data have not been previously published.

Response #3

We agree with your comment. We performed the suggested modification, and subsequently modified captions of figures and tables.

Comment #4

Figures must be improved. The use of 2 decimals and coma instead of period is unacceptable

Response #4

We agree with your comment. We overlooked this mistake, which is now fixed.

Reviewer 2 Report

The work is very interesting, but it has many limitations - which the authors mention.

Sensitivity and specificity of ultrasound in the diagnosis of liver steatosis is estimated respectively at 88%. The disadvantage of ultrasound is its limited usefulness diagnostic in low steatosis and for persons with BMI ≥40 kg/m. 

The BMI of patients was higher, which can be a serious problem for good steatosis
It is a pity that there was no additional measurement, e.g. the number of tiles that helped to calculate the Fatty Liver Index (FLI) - it is very helpful to identify people with steatosis.
Why were there no people in the exclusion criteria infection of HBV and HCV.

There is no information about the physical activity of patients before study (e.g IPAQ) - its should be measured before and in the course of the diet. 

There is no information about the current diet - it should be mesured, usually by FFQ 

How else was the adhesion to the diet checked or only with ketone bodies in urine? There is no information on what happened when the ketone bodies were absent in urine? This could be related to eating more carbohydrates.

We don't know if the diet worked, or rather the large calorie resonances.

In the introduction to the discussion, more attention was lacking for glutamil transferase (GT) This is not an enzyme normally used to assess steatosis or weight reduction.

Author Response

Comment #1

The work is very interesting, but it has many limitations - which the authors mention.

Response #1

Thank you for your favorable comment.

Comment #2

Sensitivity and specificity of ultrasound in the diagnosis of liver steatosis is estimated respectively at 88%. The disadvantage of ultrasound is its limited usefulness diagnostic in low steatosis and for persons with BMI ≥40 kg/m. 

The BMI of patients was higher, which can be a serious problem for good steatosis

Response #2

Thank you for your valuable comment. We are aware of limitations of ultrasonography in detection and classification of liver steatosis, particularly in obese subjects. This is one of the reasons we chose gGT instead of ultrasonographic degree as primary endpoint. We are now highlighting these further limitations in our discussion (Page 11, Lines 304-305).

Comment #3

It is a pity that there was no additional measurement, e.g. the number of tiles that helped to calculate the Fatty Liver Index (FLI) - it is very helpful to identify people with steatosis.

Response #3

Thank you for the valuable suggestion. Following your comment, we computed FLI in our population, but results gave us the impression of overestimating real prevalence of fatty liver. Indeed, median value of FLI before diet in our patients was 99.41% (IR 95.27-99.86). In other words, all our patients were largely above the threshold for diagnosing fatty liver (60%) proposed by Bedogni et al. After a quick search of literature, we see that FLI is not validated in subjects with severe obesity, but only in general populations of different ages and ethnicities. Likely, the high values of BMI (which is a parameter in the formula) overwhelm the effect of all other factors. In our opinion, the formula could be applicable to obese subjects with appropriate corrections, and this is a great idea for future studies. Anyhow, we are now reporting FLI in the text (Page 11, Lines 325-329).

Comment #4

Why were there no people in the exclusion criteria infection of HBV and HCV.

Response #4

These criteria were obviously included, but were accidentally not reported. We have now fixed this issue adding the following specification "chronic HCV and/or HBV infection (0 patients), and self-reported daily alcohol intake >30 g/day (0 patients)" among exclusion criteria (Page 2, Lines 71-72).

Comment #5

There is no information about the physical activity of patients before study (e.g IPAQ) - its should be measured before and in the course of the diet.

Response #5

We agree with the reviewer about the importance of accounting for daily physical activity when exploring effects of caloric restriction. Unfortunately, we did not collect this information through a specific questionnaire. In general, we thought that 25 days were a short time lapse to see measurable effects of a variation in daily physical activity (studies on physical activity are usually based on at least 6-12 weeks time lapses). Moreover, we did not expect a significant increase of physical activity during so strong caloric restriction, so we assumed that the amount of daily physical activity was non-variant or, at worse, little reduced.

Comment #6 

There is no information about the current diet - it should be measured, usually by FFQ 

Response #6

We agree with the reviewer about the importance of accounting for pre-treatment dietary habits when exploring effects of caloric restriction. Unfortunately, we did not collect this information through a specific questionnaire. During clinical history collection, we explore the dietary habits of obese patients, reporting a usual caloric intake >3000 kcal/day for males and >2500 kcal/day for females. Nonetheless, this is just an approximate evaluation, that cannot be reported in a research paper.

Comment #7

How else was the adhesion to the diet checked or only with ketone bodies in urine? There is no information on what happened when the ketone bodies were absent in urine? This could be related to eating more carbohydrates

Response #7

Adherence to diet was assessed by urinary ketone bodies. Whenever ketones should be absent, patients were excluded from the study. We are now reporting this additional exclusion criterion, which was first overlooked "Patients who did not test positive for urine ketones 96 hours after the beginning of the diet, and patients who tested negative during the treatment, were excluded by the study" (Page 4, Lines 158-159).

Comment #8

We don't know if the diet worked, or rather the large calorie resonances.

Response #8 

Thank your for your valuable comment. The argument you are raising is particularly interesting, although complex. Our opinion, as explained in the discussion section, is that caloric restriction is the key factor determining weight loss and improvement of NAFLD. Probably, described sex differences are independent of caloric restriction entity (which is the same for males and females) and probably depend on specific features of VLCKD. However, data about sex differences in other dietary treatments are lacking, so we cannot draw a definitive hypothesis on this topic. Furthermore, VLCKD allows a strong caloric restriction (<800 kcal/day) which cannot be obtained by other dietary treatments, so comparisons would be biased anyway.

Comment #9

In the introduction to the discussion, more attention was lacking for glutamil transferase (GT) This is not an enzyme normally used to assess steatosis or weight reduction.

Response #9

We agree with your comment. We have now moved the following sentence "We chose gGT as a non-invasive biomarker of NAFLD because it is a routinely performed, easily accessible, and non-expensive test. The association between gGT levels and presence and severity of NAFLD is well-established, as it is observed in our population too, and it was recently proposed as a predictor of cardiovascular risk related to NAFLD [15]" in the Introduction section.